# Classification of likely functional class for ligand binding sites identified from fragment screening
Javier S. Utgés [1], Stuart A. MacGowan [1], Callum M. Ives [1,2] & Geoffrey J. Barton [1] ✉

Fragment screening is used to identify binding sites and leads in drug discovery, but it is often unclear which binding sites are functionally important. Here, data from 37 experiments, and 1309 protein structures binding to 1601 ligands were analysed. A method to group ligands by binding sites is introduced and sites clustered according to profiles of relative solvent accessibility. This identified 293 unique ligand binding sites, grouped into four clusters (C1-4). C1 includes larger, buried, conserved, and population missense-depleted sites, enriched in known functional sites. C4 comprises smaller, accessible, divergent, missense-enriched sites, depleted in functional sites. A site in C1 is 28 times more likely to be functional than one in C4. Seventeen sites, which to the best of our knowledge are novel, in 13 proteins are identified as likely to be functionally important with examples from human tenascin and 5-aminolevulinate synthase highlighted. A multi-layer perceptron, and *K*-nearest neighbours model are presented to predict cluster labels for ligand binding sites with an accuracy of 96% and 100%, respectively, so allowing functional classification of sites for proteins not in this set. Our findings will be of interest to those studying protein-ligand interactions and developing new drugs or function modulators.

Fragment-based drug discovery or fragment screening, is widely used to identify lead compounds against a specific protein target[1]. Fragment screening typically uses X-ray crystallography to provide detailed information on the binding mode of small molecule fragments that bind to a target protein. Fragments can then be linked or grown to form more potent leads[2–4]. A typical fragment screening experiment will generate a collection of three-dimensional structures with fragments bound to different regions of the protein. While many fragments group around well understood catalytic or binding sites and so provide a scaffold for drug discovery, other fragments are also observed bound to regions of the protein where the functional significance is unclear. Such sites may be functionally irrelevant or could identify previously unknown allosteric or other functionally important sites worthy of experimental investigation.

In this paper, we describe a strategy to identify which fragment binding sites are most likely to be of functional importance and so prioritise sites for further investigation. The first step is to identify binding sites from the fragment data. We are not predicting ligand binding sites, as P2Rank[5], Fpocket[6], or molecular dynamics-based methods such as MixMD[7,8], MDmix[9], or SILCS[10] do. Instead, from a set of experimentally determined three-dimensional structures of protein–ligand complexes, we define which ligands bind to the same site, based on their protein–ligand interactions.

In most previous studies, the focus has been on clustering ligands by root-mean-square deviation (RMSD)[11] or Euclidean distances[12] after ligand superposition. Ligand site prediction resources such as 3DLigandSite[13,14] also define sites based on ligand structure superposition and RMSD. Here, we describe an algorithm that defines ligand binding sites from analysis of ligand interaction residues on the protein. The method allows the extent of a fragment binding site to be described without the need for superposition. We then apply unsupervised methods to group the defined sites into four robust clusters according to their relative solvent accessibility profiles and show which clusters are enriched in functionally characterised sites. Our analysis suggests which sites in a set of 39 fragment screening experiments are most likely to be of functional significance through further stratification by evolutionary conservation and human population missense-depletion[15,16]. We then develop a machine learning method that takes a set of interacting residues in an experimentally determined structure or a predicted ligand binding site and identifies which of the four classes best represents the site.

[1]Division of Computational Biology, School of Life Sciences, University of Dundee, Dundee, Scotland, UK. [2]Present address: Department of Chemistry and Hamilton Institute, Maynooth University, Maynooth, Ireland. ✉e-mail: gjbarton@dundee.ac.uk

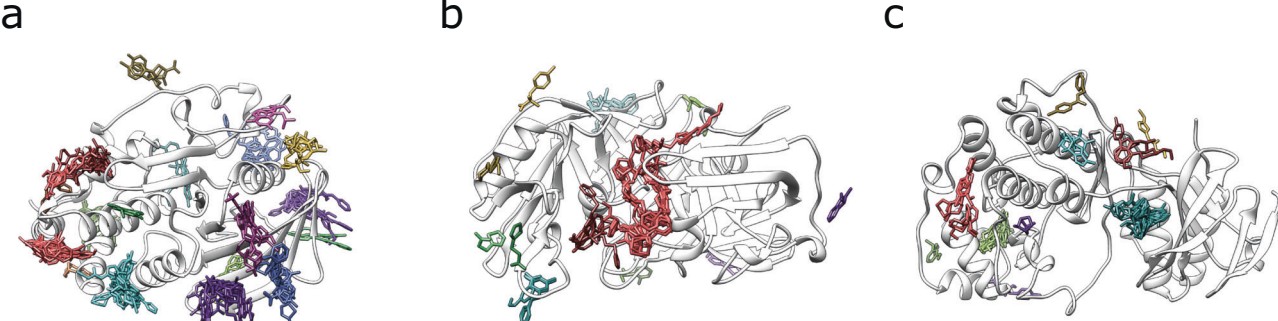

**Fig. 1 | Ligand clusters defined by the binding site definition algorithm.** For simplicity, only one protein chain ribbon is shown in white for each example. Ligands are coloured according to the site they bind to. Identifiers are from UniProt. **a** There were 110 structures depicting human tyrosine-protein phosphatase non-receptor type 1 (*PTPN1*), P18031, binding 143 ligand molecules, 104 of which were unique. 18 binding sites were defined. **b** The 68 ligands, 30 unique, found across 50 structures of the chestnut blight fungus endothiapepsin (*EAPA*), P11838, were classified in 12 distinct binding sites. **c** For mouse mitogen-activated protein kinase 14 (*Mapk14*), P47811, 52 structures portrayed the interaction with 53 ligand molecules, 50 unique, which clustered in 10 ligand binding sites.

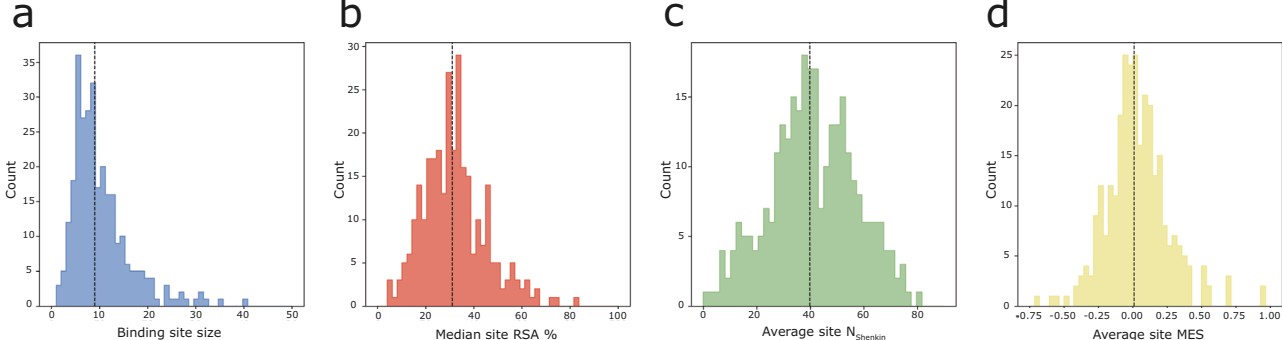

**Fig. 2 | Variation in binding site features.** Distribution of **a** size, **b** median RSA, **c** $N_{Shenkin}$ and **d** MES across the 293 binding sites defined from our dataset. Black dashed lines indicate the median of each distribution.

The work in this paper is likely to be of interest to groups focusing on fragment screening studies but wider applications to ligand site classification from experimentally determined or predicted structures are also discussed.

## Results

### Defined binding sites

The focus here is on human proteins to allow the additional information from human population variation data to be explored. For this reason, two of the 39 protein domains (products of the Replicase polyprotein 1ab from *SARS-CoV-2* (P0DTD1)) were removed since they did not include any human homologues. The remaining 37 protein domains accounted for 1309 three-dimensional structures that included interactions with 1601 ligands of interest, of which 998 were unique. 293 ligand binding sites were defined across these domains, formed by 2664 unique ligand binding residues. The total number of binding sites per domain ranges from 1–24, with 33/37 domains presenting more than one defined binding site. The median number of sites per domain is seven.

Figure 1 illustrates three examples of the 37 domains for which ligand binding sites were defined by the algorithm presented in this work. The grouping of the ligands into the defined sites reflects the similarity between the interaction fingerprints of the different ligands with the target protein domain.

Figure 2 shows the 293 defined binding sites are diverse in size (number of amino acids), solvent accessibility, evolutionary divergence, and missense depletion. Binding site size ranges from 2–40 residues with a median of 9, while median site RSA ranges from 4–80%, with a median of 30%. For evolutionary divergence, the average site $N_{Shenkin}$ spread from 0–80, peaking at 40. Lastly, MES spans −0.75 to 1.0, peaking at neutrality (MES ≈ 0).

Despite the diversity among sites, some general trends can be observed. Figure 3a shows that larger binding sites tend to be less accessible to solvent $(r = -0.4, p \approx 0)$. Figure 3b illustrates that larger sites are less divergent across homologues $(r = -0.21, p = 10^{-4})$ while Fig. 3c presents how larger sites show lower enrichment in neutral missense variants within the human population, i.e., are on average more depleted in missense variants than sites of a smaller size $(r = -0.15, p = 0.008)$. Correlations between MES and $N_{Shenkin}$, and RSA and $N_{Shenkin}$ were not significant, i.e., 95% CI $r \subset 0$.

### RSA-based binding site clustering

Figure 4a depicts the four clusters defined by our method and the RSA profiles of the sites within them while Fig. 4b illustrates six binding sites from each cluster to highlight the range of binding site size. Cluster 1 includes 46 sites, whereas 127 sites are found on C2, 91 in C3 and 29 in C4. The proportion of residues with an RSA < 25% in Fig. 4a follows a different profile in each cluster, which is confirmed in Fig. 5a. C1 is the most buried with a proportion of residues with RSA < 25% of 0.68, $(p_{RSA<25\%} \approx 0.68)$, followed by C2 with $p_{RSA<25\%} \approx 0.47$, then C3, $p_{RSA<25\%} \approx 0.30$, and lastly C4 with $p_{RSA<25\%} \approx 0.10$. Figure 5b displays the difference in binding site size between the clusters. There is variation within clusters in site size, but certain patterns are still apparent. C1 includes the largest sites, with an average size of $\bar{s} = 15$ residues, followed by C2 with $\bar{s} = 11$, then C3 with $\bar{s} = 8$, and finally C4 with $\bar{s} = 5$. Figure 5c shows the two-dimensional MDS representation of the binding sites. C1 and C4 are the most distinct amongst the clusters while there is some overlap between clusters. Sites near the cluster borders are those that switch groups depending on the random initialisation of the clustering. To summarise, C1 includes on average the largest, most buried sites, whereas C4 includes the smallest and most accessible. C2 and

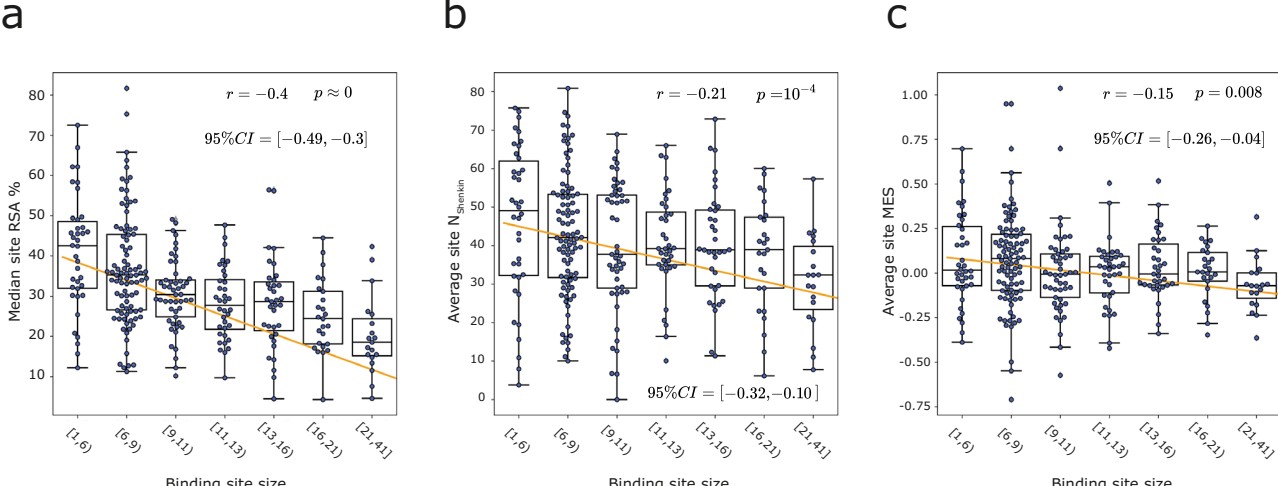

**Fig. 3 | Relation between different binding site properties.** A regression line is fitted to all data points previous to binning, ($N = 293$ binding sites), Pearson's correlation coefficient $r^{98}$, associated p-value and 95% CI of $r^{99}$. Data points are grouped into bins according to different binding site size intervals, represented by box and swarm plots. **a** Median site RSA % vs binding site size, in amino acids. **b** Average $N_{Shenkin}$ vs binding site size. **c** Average site MES vs site size. Boxes represent the IQR, and whiskers extend to $1.5 \times IQR$.

C3 are not as different as C1 and C4, but still differ in size and burial proportion with C2 including larger and overall, less accessible sites than C3.

These results support $U_D$ as a metric that effectively quantifies the difference between the solvent exposure and size properties of different binding sites with the four clusters encapsulating differences in RSA and binding site size. This effect might be explained by the negative correlation between solvent accessibility and binding site size shown in Fig. 3a.

Figure 5d shows the RSA distribution of all residues forming the binding sites within the defined clusters. This definition agrees, as expected, with Figs. 4 and 5a. C1 presents a distribution clearly different to the rest of clusters, peaking at RSA ≈ 5%, indicating a high density of buried residues. C2 still presents an excess of buried residues relative to clusters 3–4, though not as high as C1. C4 presents the most different distribution to C1, peaking around RSA ≈ 50–70%.

To further characterise the defined clusters, the distributions of the normalised Shenkin divergence score ($N_{Shenkin}$) and Missense Enrichment Score (MES) of the residues found in the clusters were analysed (Fig. 5e, f). Regarding evolutionary divergence (Fig. 5e), C1 also presents a different distribution to the rest of the clusters, with a peak at $N_{Shenkin}$ ≈ 5, i.e., most of the residues conforming the sites within this cluster are highly conserved. The other clusters present flatter distributions with increasing proportion of divergent residues ($N_{Shenkin} > 25$) $p_{C2} = 0.55$, $p_{C3} = 0.67$, and $p_{C4} = 0.69$. $N_{Shenkin}$ is a divergence score ranging from 0 to 100, therefore residues with $N_{Shenkin} < 25$, $p_{C1} = 0.58$, $p_{C2} = 0.45$, $p_{C3} = 0.33$, and $p_{C4} = 0.31$, represent stronger residue conservation, or lower divergence, than $N_{Shenkin} > 25$. This agrees with the pattern observed on the RSA distributions (Fig. 5d), as buried residues tend to be evolutionarily conserved[17,18]. In terms of missense-depletion (Fig. 5f), the distribution of C1 is slightly shifted to the left, towards more negative values, i.e., more missense-depleted residues, with $\overline{MES}_{C1} = -0.17$. The distributions of C2-4 are not statistically different, but present increasing average missense enrichment scores: $\overline{MES}_{C2} = -0.07$, $\overline{MES}_{C3} = -0.02$, and $\overline{MES}_{C4} = +0.06$. Once again, this pattern agrees with the ones observed with site size, solvent accessibility, and evolutionary divergence. Sites that are more buried tend to be bigger in size, more conserved across homologues, as well as depleted in missense variation in human.

**Clusters predict differential functional enrichment**
A key goal of this work is to identify which sites from a fragment screening experiment are most likely to be functional and so worth investigating further. Figure 6 shows the relative enrichment in functional sites across the four defined clusters. C1 is the most enriched in functional sites, with 17/46 sites being classed as of known function, ($OR = 4.46$, $p \approx 0$). C2 was next with 21/127 ($OR = 1.15$, $p = 0.75$). C3 with 6/91 is depleted relative to the other clusters, ($OR = 0.33$, $p = 0.01$), and finally C4 with 0/29, ($OR = 0.16$, $p = 0.04$). RSA-based defined clusters are differentially enriched in functional sites. Based on their enrichment, a binding site found in C1 is ≈4, ≈14, and ≈28-fold more likely to be functional than a site in C2, C3, and C4, respectively.

Functional definitions in UniProt tend to lag behind the literature. A literature search found support for 12 sites in C1 that are without UniProt annotations with two examples discussed below. We found no literature support for the remaining seventeen sites in C1 suggesting they may be novel, functionally important sites. Supplementary Table 2 shows the full list of C1 sites that are predicted to be functionally important with 2/17 examples discussed below.

**Example C1 site functional predictions supported by literature but not annotated in UniProt**
*NS3 protein from Zika virus—Q32ZE1*. The *Zika virus* (ZIKV) genome polyprotein (Q32ZE1) is 3419 amino acids long and codes for three structural proteins: capsid (C), envelope (E), and membrane (M) as well as seven non-structural proteins: (NS1, NS2A, NS2B, NS3, NS4A, NS4B, and NS5). NS3 is a critical serine proteinase for viral polyprotein processing and genomic regulation. It includes a protease domain at the N-terminus, and a helicase domain on the C-terminus. The helicase is responsible for RNA unwinding during replication, and thus makes an interesting drug target against ZIKV[19].

There are 10 sites in NS3 identified from 17 structures with 17 unique ligands and all are functionally unannotated in UniProt. The analysis here shows binding site 7 (BS7) to lie in Cluster 1 and so is most likely to be functional.

The site is located between domains I–III, involving residues from η2, α3 on domain I, and α10, α11 on domain III as defined in Tian et al.[20] (Fig. 7a). Mottin et al.[21] predicted four RNA binding sites on NS3. One of them, the RNA exit crevice is located between domains I–III, and involves α3, α10 residues. Raubenolt et al.[22] probed four different allosteric sites on this protein. One of them, D3, was manually curated and included α11, α12, and overlapped with BS7. Later, Durgam and Guruprasad[23] stated that four of the ten residues forming this site: Ala264, Thr265, Lys537 and Asp540 bind to RNA when this is in complex with NS3. These results strongly suggest that this region plays an important role in RNA binding to NS3 and

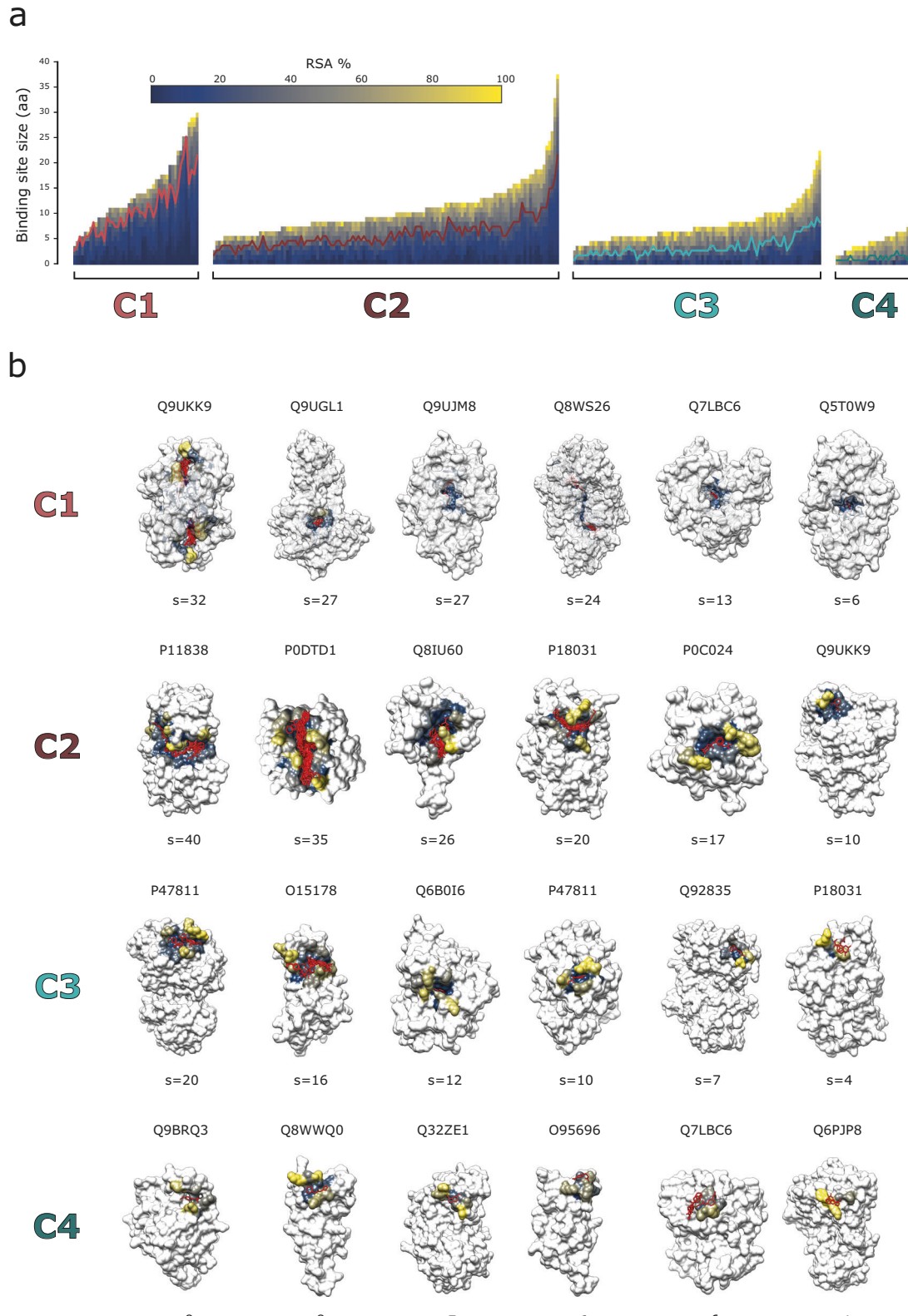

**Fig. 4 | RSA-based binding site clusters and examples. a** RSA profiles of the 293 binding sites that were grouped in four, C1-C4, clusters by *K*-means based on the difference between their RSA profiles ($U_D$). Each binding site is represented by a vector, plotted as a bar here. The elements of the vector represent the residues that form the binding site and are sorted according to their RSA, so buried residues are at the beginning of the vector (bottom), and more accessible residues towards the end (top). Each element of the vector, or section of the bar, is coloured according to RSA, using the matplotlib *cividis* colour palette. Within each cluster, binding sites are sorted based on the number of amino acids. Over each cluster, a line is drawn at RSA = 25%. **b** Six examples of binding sites are shown in structure for each cluster. Examples were selected to represent the range of binding site sizes within each cluster. IDs are UniProt accession codes. Binding site residues are coloured according to their RSA, using the *cividis* colour scheme. The rest of the protein is coloured in white. Ligands binding to the site in question are coloured in red.

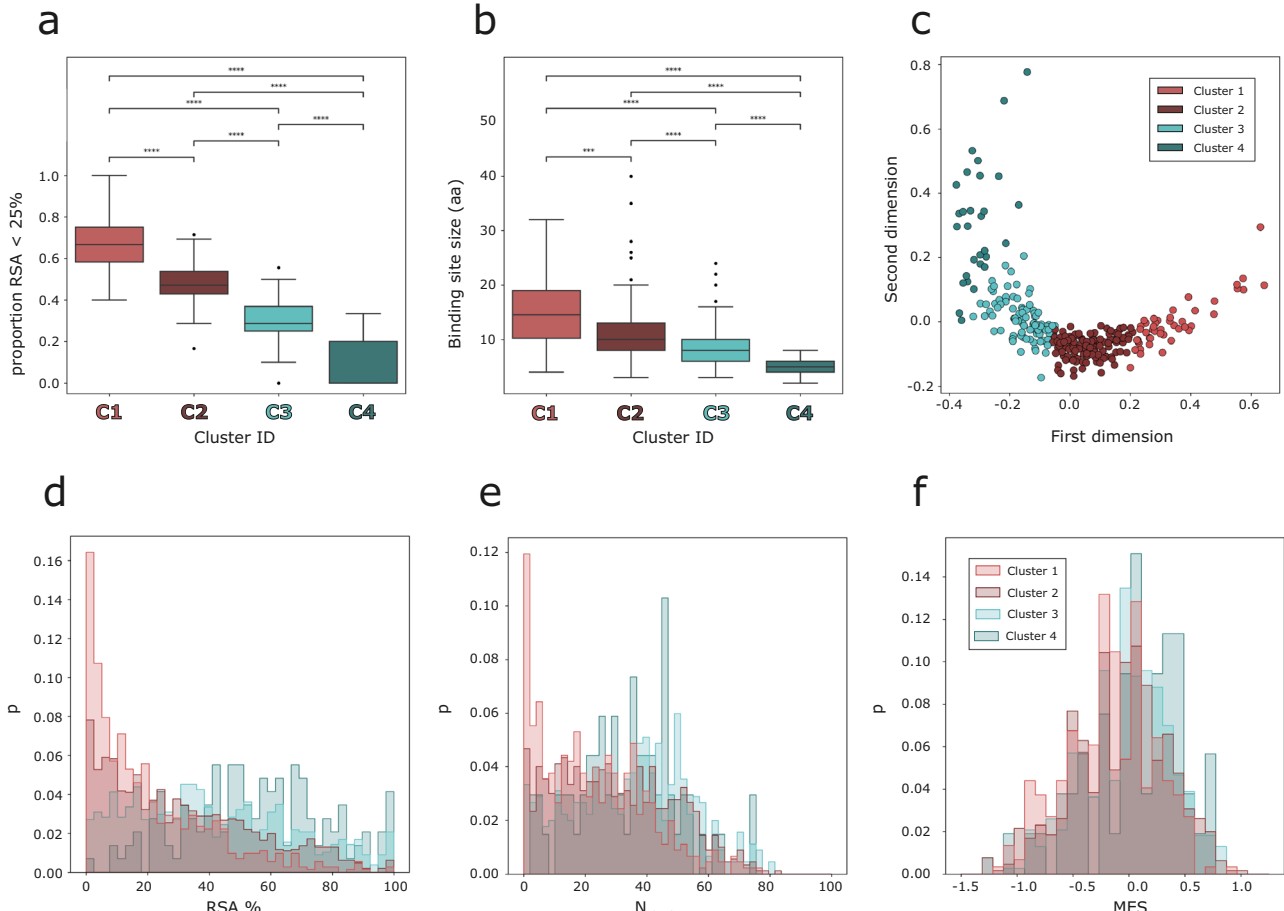

**Fig. 5 | Binding site cluster features. a** Box plot of the proportion of residues with RSA < 25% per binding site across the four clusters defined by $K$-means clustering. **b** Box plot of the binding site size, in amino acids, across clusters. Pairwise Mann–Whitney–Wilcoxon tests were performed to assess the differences between the clusters. Boxes represent the IQR, and whiskers extend to $1.5 \times$ IQR. $p$-value annotation legend: $ns : p > 0.05$, $* : 0.01 < p \le 0.05$, $** : 10^{-2} < p \le 10^{-3}$, $*** : 10^{-4} < p \le 10^{-3}$, $**** : p \le 10^{-4}$. **c** MDS representation of the 293 binding sites on 2 dimensions. Data points represent binding sites and are coloured based on the cluster they group in. **d** Histogram of RSA % of the residues found within the ligand binding sites in each cluster. **e** Histogram of $N_{Shenkin}$ within cluster residues. **f** MES histogram plots for the four clusters defined.

so is a site to target to modulate function. Moreover, the site is on average missense-depleted: MES = −0.28. A264 ($N_{Shenkin} = 18$, MES = −0.79), T267 ($N_{Shenkin} = 53$, MES = −0.55), and S293 ($N_{Shenkin} = 72$, MES = −0.48) are the three key positions out of the 10 forming this binding site, as they are all constrained within the human orthologues of this protein. A264 is conserved across homologues, whereas T267 and S293 are divergent while missense-depleted so could be important for binding specificity.

**NSP13 protein from SARS-CoV-2—P0DTD1.** The *Severe acute respiratory syndrome coronavirus 2* (*SARS-CoV-2*) replicase polyprotein 1ab (P0DTD1) is 7096 amino acids long and codes for 16 non-structural proteins[24]. NSP13 is a helicase that unwinds dsRNA in the 5'–3' direction to provide a single-stranded template for viral RNA amplification[25]. NSP13 also has NTPase activity, which provides the energy for the RNA unwinding[26]. NSP13 plays a fundamental role in the replication and transcription of the *SARS-CoV-2* genome and is thought to be a good drug target against *SARS-CoV-2* virus infection[27]. NSP13 has five domains. Two "RecA like" subdomains 1A and 2A, in charge of nucleotide binding and hydrolysis, as well as three other domains: an N-terminal zinc-binding domain, the helical "stalk" domain, and a beta-barrel 1B domain[28]. It is the most conserved protein across coronaviruses, with sequence identity >99%[29].

Twenty-four sites are defined on the surface of NSP13. Our method identifies two binding sites: BS6, and BS16 as C1 (Fig. 7b). Visual

inspection shows the two sites to be adjacent with a total of 16 residues. Three fragments bind to the site, which is located in the nucleotide and RNA binding interface of NSP13 between the 1B and 2 A domains. This is the region where the 5' end of the RNA binds[30]. This pocket is determined to be highly druggable, and drugs binding to it might be effective against other coronaviruses, due to the pocket's high amino acid conservation[31]. This agrees with our results, as this site has an average $N_{Shenkin} = 32$, and MES = −0.18. Of the 16 positions in this site, four show high conservation across homologues and missense depletion in human: P514 ($N_{Shenkin} = 30$, MES = −0.56), D534 ($N_{Shenkin} = 9$, MES = −0.56), T552 ($N_{Shenkin} = 48$, MES = −1.87), and H554 ($N_{Shenkin} = 36$, MES = −0.85). T552 shows highest conservation across species and lowest missense enrichment (−1.87) and so is most likely to have a key function in this protein family.

**Examples of potentially novel C1 cluster functional predictions**
**Human tenascin (TN)—P24821.** Human tenascin, is a hexameric extracellular matrix glycoprotein implicated in a variety of functions, including cell migration, cell attachment, matrix assembly and proinflammatory cytokine synthesis[32]. TN is known to interact with viruses and play a role in viral infections, e.g., HIV-1, and has been reported as a biomarker for disease severity[33]. It also plays a key role in wound healing[34], and is involved in diverse cardiovascular diseases[35], as well as in breast cancer[36]. For these reasons, there is considerable effort put into

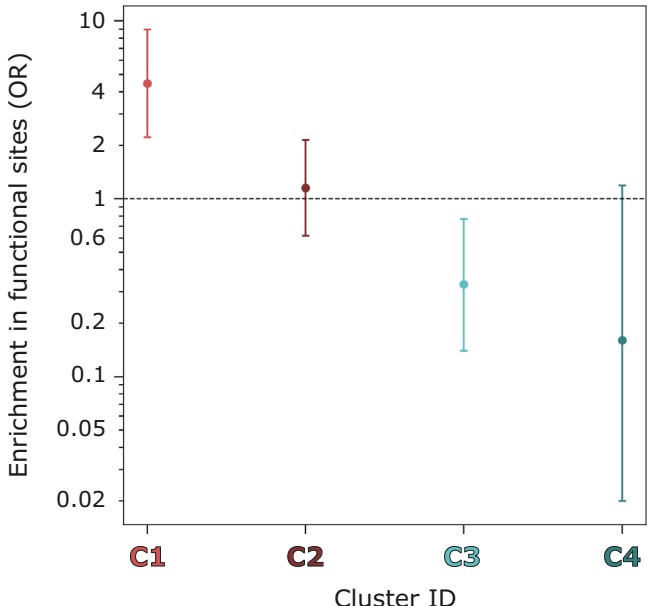

**Fig. 6 | Binding site cluster enrichment in known functional sites.** This enrichment score is an odds ratio (OR). Error bars indicate 95% CI of the OR. *Y* axis is in $\log_{10}$ scale. A pseudo-count of 1 was added to each cell of the contingency table to calculate the score.

understanding better the function of TN and targeting it for therapeutic effect.

The data includes 11 structures with 11 unique ligands binding to TN, grouped in four binding sites, none annotated as functional in UniProt. One of the four binding sites is in C1, and so predicted to be of functional importance. The site is found on the Fibrinogen C-terminal domain of the protein, which functions as a molecular recognition unit that interacts with either proteins or carbohydrates (Fig. 7c). This site shows high conservation across species ($N_{Shenkin} = 15$), and is missense-depleted in human (MES = $-0.33$). Accordingly, we suggest that this site in TN is likely to be of key importance to function. Within the 15 positions forming the site, V2012 ($N_{Shenkin} = 5$, MES = $-1.0$), G2046 ($N_{Shenkin} = 0$, MES = $-0.67$), F2047 ($N_{Shenkin} = 0$, MES = $-0.67$), W2055 ($N_{Shenkin} = 0$, MES = $-0.54$), and G2057 ($N_{Shenkin} = 0$, MES = $-0.83$), are the most critical interacting residues, and extremely conserved across homologues.

**5-aminolevulinate synthase (ALAS-E)—P22557.** *ALAS2* is a gene located on the X chromosome that codes for the human mitochondrial erythroid-specific 5-aminolevulinate synthase. This dimeric enzyme carries out the first and rate-limiting step of the haem synthesis pathway: the pyridoxal 5'-phosphate (PLP)-dependent condensation of succinyl-CoA and glycine to form aminolaevulinic acid[37]. Across eukaryotes, these enzymes have developed extensions surrounding the catalytic core on both the N and C-termini[38]. The N-terminal extensions include the mitochondrial targeting sequence[39], whereas the C-terminal extension (C-ext) plays an autoinhibitory role by regulating substrate binding and product release[40]. Mutations affecting C-ext can result in gain-of-function, such as X-linked protoporphyria[41], as well as loss-of-function disorders, e.g., X-linked sideroblastic anaemia[42]. Accordingly, ALAS-E is a potential therapeutic target for the treatment of such diseases.

We considered 25 structures with 33 unique ligands binding to ALAS-E, grouped in ten binding sites, only one of which is annotated as functional in UniProt. We classify three sites as C1. Two are known to be on the interface between subunits, form key interactions to maintain the assembly and are close to the PLP binding site[40]. However, one (BS1) is not mentioned in the literature. This site is located on a deep pocket at the N-terminal

region of the protein structure (Fig. 7d). Amino acids in this site are strongly conserved as well as depleted in missense variation: $N_{Shenkin} = 29$, MES = $-0.13$. Together, this suggests the site has a functional role in the protein, perhaps as an allosteric regulator, or through interaction with a partner such as succinate-CoA ligase, SCS-$\alpha$[43]. Out of the 16 residues forming the site, K381 ($N_{Shenkin} = 38$, MES = $-0.94$) is the most missense-depleted position in the site and should be considered for lead optimisation of a fragment binding to this site.

## Discussion

In this paper, we have presented a method to identify binding sites from fragment screening data and group the sites into four robust clusters by an RSA profile metric. 29/46 sites in Cluster 1 have functional support from the literature (UniProt 17/46; Our search 12/46 — Supplementary Table 2). 17 further sites have similar profiles, but we could not find evidence in the literature of functional significance. We show two examples from this set that have compelling support from conservation and missense-depletion scores for functional significance and we list all sites in Supplementary Table 2 as a resource for further experimentation on these proteins.

As a case study, we applied the method to the *SARS-CoV-2* main protease, MPro (P0DTD1). Twenty-five sites were defined from 511 structures, from which 8 were classed as C1, 12 as C2, 3 as C3 and only 2 as C4. Of the 8 C1 sites, one corresponds to the active site and three to allosteric sites 1, 2, and 3, as defined by DasGupta et al.[44], respectively. A further C1 site is at the dimer interface and known to be a potential allosteric site[45] (see Supplementary Fig. 4). The remaining three C1 sites may be important, but each binds only a single ligand and their function is currently unclear.

Here, we have focused on a small set of proteins heavily studied by fragment screening methods. However, our method can be applied to classify any ligand binding site or predicted site. Accordingly, future work will seek to classify all known ligand binding sites in the PDB and provide tools to predict likely functional sites predicted sites by tools such as P2Rank[5], or GRaSP[46] from Alphafold2[47,48] or other models.

It is natural to focus on sites that are most likely to be of functional significance and so possible targets to modulate function. However, binding sites identified here that are predicted to be least likely to have function may also be interesting as good locations for tagging proteins for degradation[49], phosphorylation[50], dephosphorylation[51], or other modulation[52,53].

## Methods
### Structure dataset
The Pan-Dataset Density Analysis (PanDDA) algorithm characterises a set of related crystallographic datasets of the same crystal form and identifies binding events by isomorphous difference maps[54]. Initially, 3021 three-dimensional structures determined by X-ray crystallography were selected by querying the PDBe[55] for entries containing the string "PanDDA" in their title. In total, 1542 of the structures included bound ligands for 39 different proteins. Four proteins which were in multiprotein complexes including additional ligands were excluded to leave protein–ligand complexes coming from 35 different proteins and a total of 1450 three-dimensional structures. The structures presented resolutions from 0.9 to 3.3 Å, with a mean resolution of ≈1.5 Å. The preferred biological assemblies, as defined by PISA[56], were downloaded from the PDBe via ProIntVar[57].

### Binding site definition
Ligand binding site definition or prediction approaches are usually based on the spatial superposition and clustering of the atomic coordinates of ligands according to Euclidean distances or RMSD[11–14]. These methods rely on structural superposition but can be computationally expensive when dealing with large numbers of structures. Here, we define sites from protein–ligand interactions without the need for superposition (Fig. 8). Only non-ion ligands of interest were used for the binding site definition. These do not include water molecules, nor other by-products of the

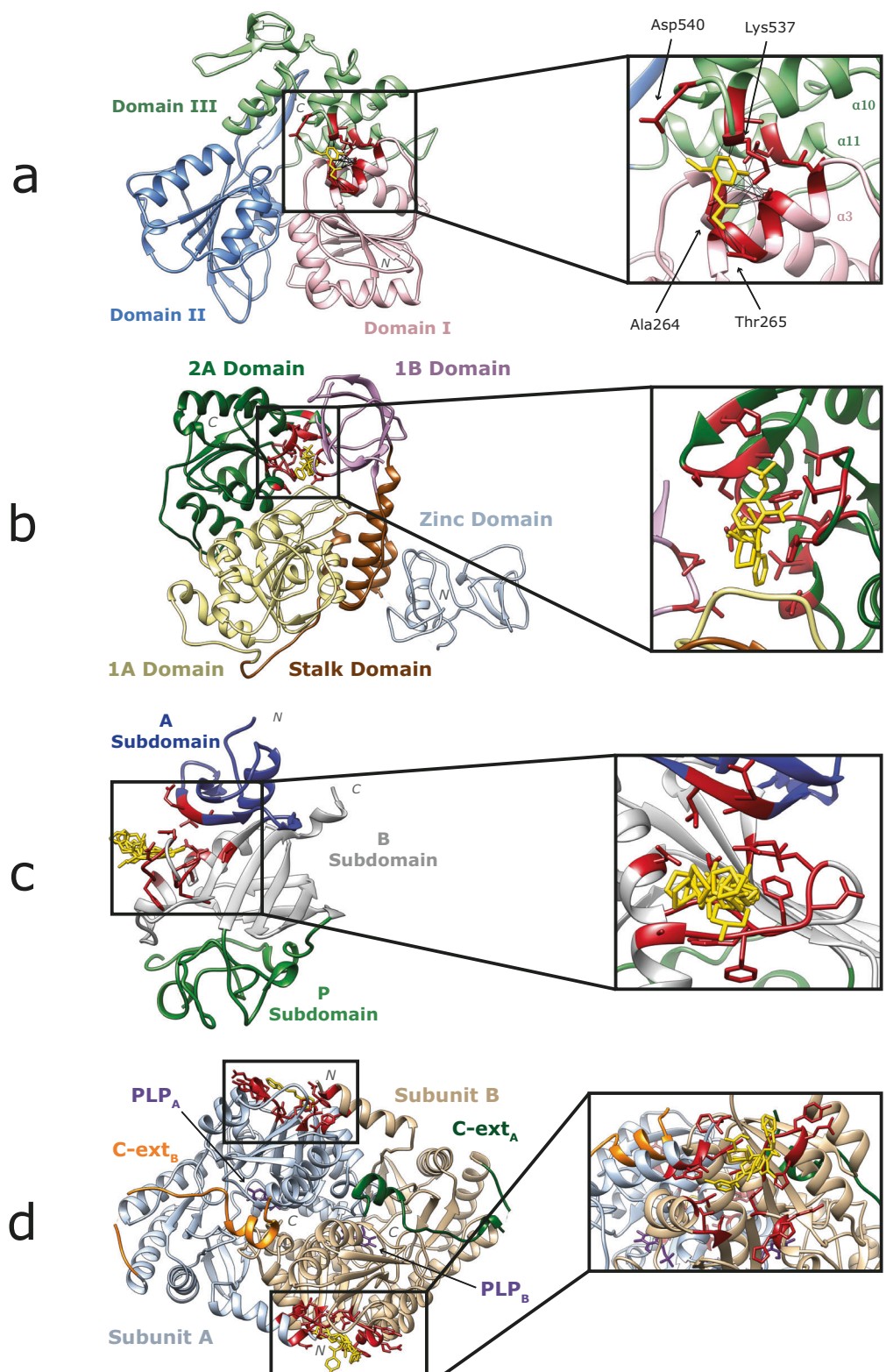

**Fig. 7 | Examples of C1 sites of interest. a** Non-structural protein NS3 of *Zika virus* (Q32ZE1) binding to N-(2-methoxy-5-methylphenyl)glycinamide, NY7 in BS7 (PDB: 5RHG) (Godoy AS, Mesquita NCMR, Oliva G). Domains I, II, and III are coloured in pink, blue, and green, respectively. Binding site 7 which is in Cluster 1 is highlighted, the other 9 binding sites which fall in C2 (3), C3 (3) and C4 (3) are hidden. Ligand binding residues in red, and NY7 in yellow. Protein–ligand interactions are represented by black lines. **b** Non-structural protein NSP13 of *SARS-CoV-2* (P0DTD1) binding to three ligands in BS6 + 16 (Ribbon PDB: 5RMH)[31]. 1A, 1B, 2A, stalk and zinc domains are coloured in yellow, pink, green, brown, and grey, respectively. Ligand binding residues in red, and ligands in yellow. Interactions are not shown here for simplicity. **c** Human tenascin, TN, (P24821) binding to 8 ligands in BS0. (Ribbon PDB: 5R60) (Coker JA, Bezerra GA, von Delft F, Arrowsmith CH, Bountra C, Edwards AM, Yue WW, Marsden BD). A, B, and P subdomains as defined by Yee et al.[100] are coloured in blue, grey, and green, respectively. **d** Human erythroid-specific mitochondrial 5-aminolevulinate synthase, ALAS-E, (P22557) binding to 7 ligands in BS1. (Ribbon PDB: 5QR0) (Bezerra GA, Foster W, Bailey H, Shrestha L, Krojer T, Talon R, Brandao-Neto J, Douangamath A, Nicola BB, von Delft F, Arrowsmith CH, Edwards A, Bountra C, Brennan PE, Yue WW). Subunits A, B, C-terminal extensions A, B, as well as PLP cofactors are coloured in grey, beige, green, orange, and purple, respectively. Ligand binding residues in red, and ligands in yellow.

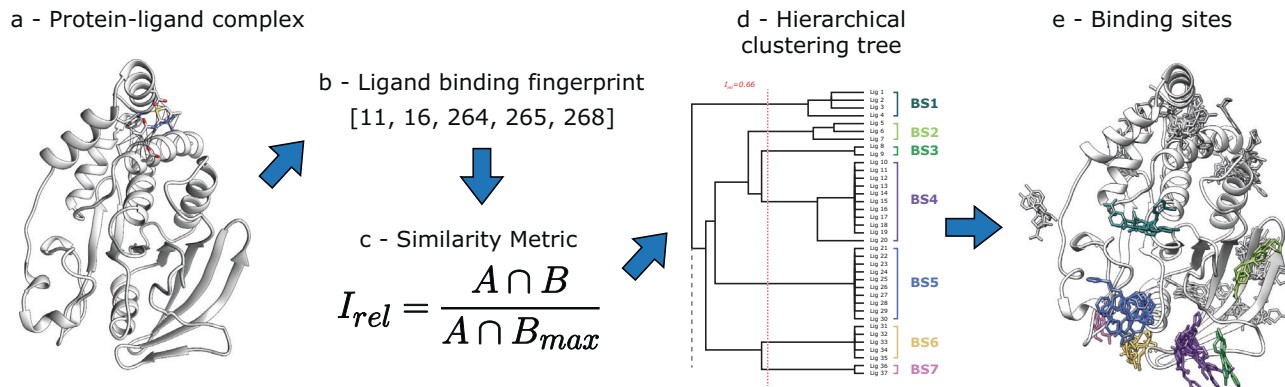

**Fig. 8 | Ligand binding site definition algorithm.** The method defines ligand binding sites from a set of three-dimensional structures portraying the complex of a protein of interest bound to ligands. **a** Protein–ligand complex (P18031). **b** Ligand binding fingerprint, comprised by protein residue numbers interacting with ligand. **c** Formula of the similarity metric: relative intersection, $I_{rel}$. **d** Hierarchical clustering tree resulting from the similarity matrix, cut at threshold to determine distinct clusters of ligands. **e** Three-dimensional structure of all ligands binding to protein, coloured according to the cluster they group into. Only ligands found clusters 1–7 are in coloured based on their cluster membership. The rest are coloured in grey. The tree on (**d**) represents only a part of the tree, showing 7/18 binding sites defined on P18031. This is represented by a dash line pointing downwards on the tree.

experimental conditions. Ligand contacts were determined with Arpeggio[58]. For a given ligand, a binding fingerprint is defined as the UniProt residue numbers the ligand interacts with. For a pair of ligands $L_A$ and $L_B$, with their interaction fingerprints A and B, their relative intersection, $I_{rel}$, is defined (Eq. 1) by dividing the intersection of sets A and B by the maximum possible intersection between the two sets, given by the minimum fingerprint length (Eq. 2). $I_{rel}$ ranges from 0–1.

$$I_{rel} = \frac{A \cap B}{A \cap B_{\max}} \qquad (1)$$

$$A \cap B_{\max} = \min(len(A), len(B)) \qquad (2)$$

$I_{rel}$ is thus a similarity metric that can be used to perform hierarchical clustering on the ligands. Single-linkage hierarchical clustering was performed with the OC software[59]. After exploring several threshold $I_{rel}$ values to cut the resulting tree, we settled on $I_{rel} = 0.66$. Since this is a similarity metric, it means that a ligand shares at least two thirds of its binding residues with at least one other member of the same cluster. A total of 293 ligand binding sites across 37 protein domains were defined this way. For each protein, all structures were multiply aligned by STAMP[60]. Ligand binding sites were visualised in UCSF Chimera[61].

**Multiple sequence alignments**
Two of the 35 proteins included fragment screening experiments targeting multiple domains, or protein products, resulting in 39 protein-fragments sets. A representative sequence was selected for each of the 39 sets of structures, and used to search SwissProt[62] for homologues with jackHMMER[63] with default parameters and 5 iterations to generate multiple sequence alignments. Evolutionary divergence within the alignments was quantified with the Shenkin divergence score, $V_{Shenkin}$[64], and the normalised $N_{Shenkin}$, as defined in Utgés et al.[65].

**Human variants and enrichment**
VarAlign[57] was used to retrieve genetic variants from gnomAD v2.1[66] found in the human sequences within the multiple sequence alignment generated for each target protein. gnomAD contains exomes and genomes of 141,456 unrelated individuals with no known phenotypic conditions and is therefore a reasonable representation of the general healthy population. Variants found in the human sequences within the alignments were mapped to individual alignment columns and missense enrichment scores (MES) were calculated. MES represents the enrichment in missense variants of an alignment column relative to the average of the other columns in the alignment[15]. 95% confidence intervals (CI) and $p$-values were used to assess the significance of these ratios[67]. MES was also calculated for the defined ligand binding sites. The MES of a binding site represents the enrichment in missense variants of a binding site relative to the rest of protein residues. Alignment columns as well as binding sites were classified as enriched (MES > 0), depleted (MES < 0) or neutral (MES = 0). Enrichment was not calculated for two of the 39 proteins since no human homologues were identified.

**Binding site clustering**
Secondary structures were defined with DSSP[68] via ProIntVar, and relative solvent accessibility (RSA) was calculated with the method of Tien et al.[69]. The defined binding sites were grouped according to the pattern of RSA as follows and summarised in Fig. 9.

Given two binding sites, A and B, with RSA profiles $r_A$ and $r_B$ and sizes $n_A$ and $n_B$, respectively, in amino acid residues, $U_A$ and $U_B$ can be calculated (Eq. 3). The Mann–Whitney U statistic[70], as implemented in SciPy[71], was chosen as it has a maximum theoretical value ($U_{max}$) (Eq. 4). A relative U value, $U_{rel}$, ranging 0-1 is obtained by dividing the U value by $U_{max}$. The more similar $r_A$ and $r_B$ are, the bigger $U$ and $U_{rel}$ are. Thus, $U_{rel}$ is a similarity score. Subtracting $U_{rel}$ from 1 gives the U distance, $U_D$, (Eq. 5). $U_D$ is indicative of how different $r_A$ and $r_B$ are and can be used to cluster binding sites according to their RSA profiles.

$$U_A = R_A - \frac{n_A(n_A + 1)}{2}, U_B = R_B - \frac{n_B(n_B + 1)}{2} \qquad (3)$$

$$U_A + U_B = n_A n_B, U = \min(U_A, U_B) \rightarrow U_{max} = \frac{n_A n_B}{2} \qquad (4)$$

$$U_{rel} = \frac{U}{U_{max}} \rightarrow U_D = 1 - U_{rel} \qquad (5)$$

After calculating pairwise distances between the RSA profiles of the defined binding sites, K-means clustering[72] was performed. Several clustering algorithms were tried to realise this task, including some hierarchical, or connectivity-based, such as single and complete-linkage[73], unweighted average linkage clustering (UPGMA)[74], or Ward linkage[75], as well as centroid-based, such as K-means. Overall, the clusters obtained by the different methods were similar. Ward linkage and K-means resulted in the most similar clusters, displaying an average similarity between clusters of 85% (see Supplementary Fig. 1). Finally, multidimensional

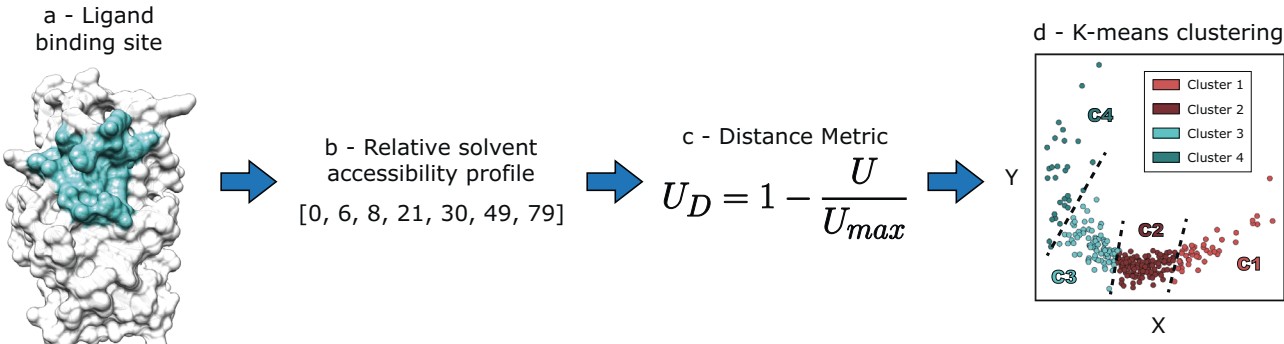

**Fig. 9 | Binding site clustering algorithm.** The method here clusters ligand binding sites defined across different proteins based on their solvent accessibility profiles. **a** Example of a defined ligand binding site. **b** Relative solvent accessibility profile of a binding site, represented by the RSA of the site residues. **c** Formula of our distance metric: distance U, $U_D$. **d** Multidimensional scaling (MDS) representation of binding sites coloured according to the four clusters determined by the $K$-means algorithm. Dashed lines represent the cluster limits.

scaling (MDS)[76] with $N = 2$ dimensions was performed to visualise the clusters. We settled on $K$-means, as it presented better contained clusters, i.e., less overlapping between members of distinct clusters. The silhouette[77], elbow[78], as well as Calinski–Harabasz index (CHI)[79] and Davies-Bouldin index (DBI)[80] methods were used for finding optimal $K$ (see Supplementary Fig. 2), in conjunction with the MDS, trees resulting from hierarchical clustering algorithms, and the visual representation of the RSA profiles, to decide on a final number of $K = 4$ clusters: C1, C2, C3, and C4. Clustering was repeated 1000 times with different random states and 289/293 (98.6%) sites were always present in the same cluster, thus suggesting the clusters are robust.

**Binding site cluster prediction**

Two different predictive models were developed with the aim of classifying binding sites into the defined RSA-based clusters obtained with $K$-means, as described above. The first uses the $K$-nearest neighbour (KNN) algorithm as implemented in Scikit-learn[81], with $K = 3$. The input for this KNN model is the rows of the $U_D$ matrix, containing the distances between pairs of binding site RSA profiles.

The second model is a multilayer perceptron (MLP)[82], a type of artificial neural network (ANN) constructed with Keras[83] with a single hidden fully connected layer between the input layer of 11 neurons, and the output layer of 4 neurons, one for each cluster label. RSA profiles present different lengths depending on the size (number of amino acids) of the binding site. As this input is not suitable for the neural network, binding sites were encoded as an 11-element vector. The first element of the vector encodes the size of the binding site relative to the maximum site size of 40 residues. The other 10 elements represent the proportion of residues forming the binding site with an RSA % within a 10-unit interval: [0, 10), [10, 20), …, and [90, 100]. In developing the method, we explored the hyperparameter space including number of hidden layers, neurons per layer, activation and loss functions, weight initialisers and optimisers (see Supplementary Note 1, Supplementary Fig. 3 and Supplementary Table 1).

The complete dataset ($N = 293$) was split into a blind test set (1/11 = 27), and a training set (10/11 = 266). Ten repeats of a stratified 10-fold cross-validation were performed to assess the robustness of the ANN and compare it with the KNN model, as well as a baseline of the same models trained on randomly shuffled data and completely random label assignment ($p = 0.25$). The reliability of the ANN predictions was assessed by means of a confidence score calculated as in Cuff and Barton[84], which represents how certain the MLP is of each individual prediction (Eq. 6). The score is based on the difference between the top- and second-class probabilities assigned by the network to each of the classes, $p_1$, and $p_2$, respectively. For example, if the output of the network were $p = [0.95, 0.02, 0.03, 0.0]$. The probabilities would be sorted,

so $p_1 = 0.95$, $p_2 = 0.03$, and a confidence score of 9 would be obtained.

$$confidence\ score = \lfloor 10 \times (p_1 - p_2) \rfloor \qquad (6)$$

The KNN is based on distances to all training data and so, as expected, consistently gives higher classification accuracy than the ANN model where sites are represented by their binned RSA profile, and are thus completely unaware of other sites, and their distances to them (Fig. 10a). Both methods are significantly better than random. The average cross-validation accuracy across all repeats is of 98%, 90%, 33%, 31%, and 24% for KNN, ANN models, their randomly trained versions, and completely random label assignment, respectively. The baseline accuracy of the randomly trained models is higher than 25% since the dataset is unbalanced, with classes, C1 and C2 overrepresented.

Figure 10b shows the confidence of the ANN predictions across the 10 repeats of the 10-fold cross-validation. The overall accuracy is 90%. Those predictions presenting a confidence score greater or equal to 5 present an accuracy of 97% and cover 75% of all predictions. Finally, Fig. 10c shows the same two-dimensional representation of the $K$-means clusters found on Fig. 5c and demonstrates that those binding sites with lower prediction confidence are mostly located at the borders between clusters. Sites that switch cluster labels depending on the seed are also located in these regions.

Once the model hyperparameters were optimised, 50 models were trained on 10/11 of the data ($N = 266$) for the ten different seeds used to initialise the models. From a final pool of 500 models, the one presenting the highest validation accuracy and lowest validation loss was chosen, with a validation accuracy of 96%. This model, as well as KNN were used to predict on the blind test set. There is no significant difference in performance of the ANN and KNN models. Accuracies are 26/27 = 0.96, 95% CI = [0.82, 0.99], and 27/27 = 1.0, 95% CI = [0.88, 1.0], for ANN and KNN, respectively. The adjusted Rand index (ARI)[85,86], as well as adjusted mutual information (AMI)[87,88] were calculated. $ARI_{ANN} = 0.93$, 95% CI = [0.81, 1.0][89], $AMI_{ANN} = 0.93$, 95% CI = [0.82, 1.0]. $ARI_{KNN} = 1.0$, $AMI_{KNN} = 1.0$. 95% CI of AMI was calculated by bootstrap resampling ($N = 10,000$). The three metrics all agree on the high performance of the MLP. Figure 10c illustrates how the binding site, which label was wrongly predicted by the ANN model is located on the limits between adjacent clusters C3 and C4. This result agrees with the $K$-means clustering reliability, and confidence score analysis of the cross-validation, where the same inter-cluster regions are highlighted due to their lower clustering reliability, and low confidence prediction. This suggests that the core of the clusters is stable, and that the ANN confidence score may be used to identify binding sites that are at the borders of clusters.

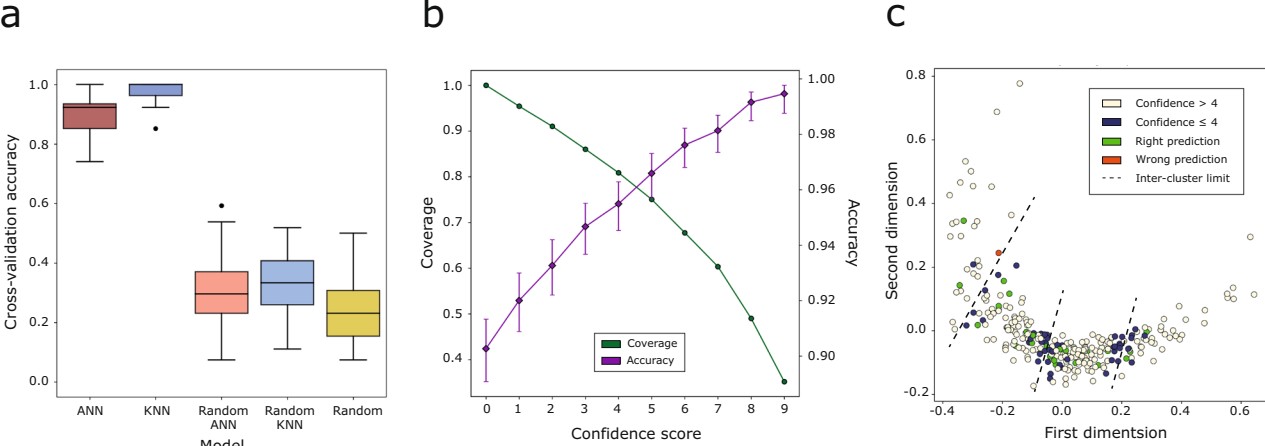

**Fig. 10 | MLP cross-validation and blind test results. a** Average accuracy of the 10-repeat 10-fold ($N = 100$) cross-validation of the KNN, and ANN predictive models compared to a baseline of the same models trained on randomly shuffled data, as well as complete random prediction ($p = 0.25$). The box represents the central 50% of the data, i.e., Q1 — median (Q2) — Q3, also known as interquartile range (IQR). Whiskers extend to $1.5 \times$ IQR, and beyond them are the outliers. **b** Cross-validation accuracy and proportion of binding sites against cumulative confidence score from the trained ANN. Sites presenting a confidence score greater or equal to 5, the average accuracy is 97%, and the percentage of sites with this score is 75%. Predictions are for the 2660 cross-validation data points, 10 different repeats of 10 distinct splits of 26–27 binding sites each. Accuracy error bars indicate 95% CI of the proportion[101]. **c** MDS representation of the 293 binding sites. Training data are coloured according to the average confidence of their prediction in the cross-validation. Test data are coloured according to whether they were correctly predicted or not. Dashed lines indicate the limits of *K*-means clusters.

## Site function classification
Ligand binding sites were divided into two groups *known function* and *unknown function* by searching UniProt[90] for feature annotations indicative of function, e.g., metal, substrate binding, active site, etc *via* the UniProt proteins API[91]. Seventeen out of the 35 proteins presented at least one UniProt annotated residue in one binding site. Manual curation using protein homology within the proteins in the dataset added 9 more functionally annotated proteins. This gave a total of 44 sites from 26 proteins classified as of known function. All other sites were classified as unknown function.

## Statistics and reproducibility
All data analysis was carried out primarily with the following Python libraries: NumPy[92], Pandas[93,94] and SciPy. Keras and Scikit-learn were used for machine learning, with Matplotlib[95], and Seaborn[96] for plotting. All statistical tests performed are two-tailed, and significance level $\alpha = 0.05$. Sample sizes and measures of significance are reported and described in the text, figures and legends.

## Reporting summary
Further information on research design is available in the Nature Portfolio Reporting Summary linked to this article.

## Data availability
The main summary result tables resulting from this analysis are available in the following repository: https://github.com/bartongroup/FRAGSYS (https://doi.org/10.5281/zenodo.10606595)[97].

## Code availability
Software developed to carry out this analysis is also found in our GitHub repository: https://github.com/bartongroup/FRAGSYS (https://doi.org/10.5281/zenodo.10606595).

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

## Acknowledgements

We thank Drs. Marek Gierliński, and James Abbott for their insightful suggestions. We also thank the IT service of the University of Dundee for their support of the HPC infrastructure this study was carried out on. We thank a referee for suggesting the MPro case study that highlights the utility of our work in that system. This work was supported by grants to G.J.B. from UKRI-Biotechnology and Biological Sciences [BB/J019364/1; BB/R014752/1] and Wellcome Trust [101651/Z/13/Z; 218259/Z/19/Z]. J.S.U. was supported by a BBSRC EASTBIO Ph.D. Studentship [BB/J01446X/1].

## Author contributions

G.J.B., J.S.U., S.A.M. and C.M.I. conceived, designed, and developed the research. J.S.U. and C.M.I. analysed the data. J.S.U., C.M.I., and S.A.M. developed the software. J.S.U. and G.J.B. wrote the manuscript. J.S.U. and G.J.B. reviewed and edited the manuscript. G.J.B. secured funding and supervised.

## Competing interests

The authors declare no competing interests.
