## [Peer Review File · Communications Biology]

Reviewers' comments:

Reviewer #1 (Remarks to the Author):

The present work describes a novel algorithm to identify likely functional sites in protein structures, derived from a machine-learning approach using PanDDA complexes as base dataset.

The authors devised a rather interesting metric to cluster the several experimentally-determined independent binding pockets.

Overall, the text is very well written and the methodology is robust. Moreover, the subject is of great interest for the scientific community.

That being said, here are some minor suggestions I believe would be beneficial for the final work:

1. Title: the term "functional state" implies some sort of dynamical change is being evaluated. I suggest refraining from using it.
2. P7L169: It is not clear how the authors reached the number of 293 binding sites, I suggest this amount is explicitly mentioned in the "Binding site definition" section
3. P7L178: Should the confidence score read 9.2?
4. P8L182: Remove comma after "accuracy".
5. P8L202: Here, the superimposed colors of ANN and "ANN random" are too similar. Moreover, this representation hinders the interpretation of the whiskers.
6. P11L272: I suggest clarifying the expression "more depleted" as its meaning is not clear here.
7. P11L280: ME  MES
8. P15L337: I suggest adding a reference for the statement "buried residues tend to be evolutionarily conserved"
9. P16L363: Are there not seventeen remaining sites?
10. P17L380: Remove first occurrence of the word "identified"

Reviewer #2 (Remarks to the Author):

Classification of likely functional state for ligand binding sites identified from fragment screening

General Assessment of work:

This is an appreciable effort to design a methodology to cluster ligands by binding sites using machine learning techniques. The authors' efforts are praiseworthy, and their study has the promise to offer a valuable addition to the field of studying drug target interactions.

Comments for revision:

1. The abstract should begin with an introductory sentence to provide context and highlight the importance of the research. Additionally, the sentence mentioning "An artificial neural network and a K-nearest..." is vague. It's worth noting that the experiment actually involved a multi-layer perceptron (MLP). Please be concise.

2. The problem definition lacks clarity. When reading the abstract, it initially appears to be addressing an unsupervised clustering problem. However, the abstract concludes by mentioning accuracy as an evaluation metric, which is not typically applicable in clustering scenarios. Clustering quality is more accurately assessed using metrics like inertia, silhouette score, Davies-Bouldin index, Calinski-Harabasz index, Dunn index, Rand index, adjusted Rand index, normalized mutual information, and others specifically designed for this purpose. Interestingly, none of these metrics are mentioned in the results section. Again, when we read about the design of ANN model in the "Binding site cluster prediction" section, it seems like this is a supervised classification problem, given that the output layer consists of 4 output nodes (one for each cluster label). Things are not organized properly, please be clear.

3. Relying solely on Accuracy is an inadequate metric for assessment; it is advisable to incorporate additional metrics for a more comprehensive evaluation of your work.

4. In the 'Binding site definition' section, provide a comprehensive overview of drug-target interactions, considering that factors such as binding affinity and binding pose (structural characteristics) are equally crucial for comprehending the binding site. You may refer to the referenced paper for guidance : (<https://doi.org/10.1093/bib/bbab476>).

5. I am skeptical about the phrase: 'These results suggest that the KNN is better than the ANN at predicting RSA-based cluster labels' in the section 'Predicting RSA cluster membership'. It's worth noting that the 'ANN' mentioned in this context employs only a single hidden layer, making it the most

basic form of a neural network. Therefore, it would not be appropriate to draw a conclusion about KNN's superiority over ANN without conducting any ablation studies or further analysis.

6. Continuing the comment #5, Authors need to perform various ablation studies as mentioned below:

a) Layer removal: Remove one or more layers (e.g., hidden layers) from the MLP and observe how it affects the model's performance. This helps assess the necessity of each layer in the network.

b) Node Removal: Experiment by selectively removing neurons (nodes) from one or more layers to evaluate the impact on model performance.

c) Activation Function Ablation: Assess the effect of using different activation functions in the MLP. For example, you can compare results when using ReLU, sigmoid, or tanh activation functions in various layers.

d) Weight Initialization: Investigate the influence of different weight initialization methods, such as random initialization, Xavier/Glorot initialization, or He initialization. Ablate certain weight initialization techniques to see how they affect convergence and model performance.

e) Hyperparameter Ablation: Systematically change and ablate various hyperparameters like the number of hidden units, the number of layers, or the dropout rate to understand their impact on the model's performance.

f) Loss Function Variations: Experiment with different loss functions (e.g., mean squared error, cross-entropy) to see how they affect the training and performance of the MLP.

Additionally, you have the option to conduct ablation experiments involving regularization techniques, learning rates, and optimization algorithms. Create a table to document all the outcomes and subsequently compare these results with those from the KNN approach.

7. Ensure that the readme documentation in the GitHub repository is comprehensive and clear enough to guide anyone attempting to replicate the results. Presently, the instructions for reproducing the results are insufficient.

Reviewer #3 (Remarks to the Author):

The authors discuss a novel method to elucidate binding site identification. There are several fragmentation/solvent based MD simulations (MDMix, MixMD, SILCS) that have successfully been used for site IDing. The authors should discuss the validity of these methods and preface them in the INTRO Section.

What makes this method unique should be clearly articulated.

Thirdly the authors must compare their data to at least freely accessible FTMap tool and analyse how the results differ from their method versus FTMap, and if any analyses could be shown to prove their findings.

The functional validation of sites is an interesting concept, it would be useful if the authors could perform a quick case study on MPro (the main protease for COVID-19) which has 2 allosteric sites and then show how their method can predict the allosteric communication. Please see paper <https://pubmed.ncbi.nlm.nih.gov/35107014/> and try to analyse if these allosteric sites are indeed predicted through this method.

Response to reviewers'

We thank all three reviewers for their insightful comments and suggestions. We answer the comments below and have made substantial changes to the manuscript to incorporate their ideas as indicated.

We feel that the referee suggestions improve our manuscript and so hope that it is now acceptable for publication.

Reviewers' comments

Reviewer #1 (Remarks to the Author):

The present work describes a novel algorithm to identify likely functional sites in protein structures, derived from a machine-learning approach using PanDDA complexes as base dataset.

The authors devised a rather interesting metric to cluster the several experimentally determined independent binding pockets.

Overall, the text is very well written, and the methodology is robust. Moreover, the subject is of great interest for the scientific community.

That being said, here are some minor suggestions I believe would be beneficial for the final work:

1. Title: the term "functional state" implies some sort of dynamical change is being evaluated. I suggest refraining from using it.

"State" is a generally used term that does not always refer to its special use in dynamics. However, we understand this may confuse some readers and so have changed the word "state" to "class" in the title which should avoid this ambiguity.

2. P7L169: It is not clear how the authors reached the number of 293 binding sites; I suggest this amount is explicitly mentioned in the "Binding site definition" section.

Text has been edited in the "Binding site definition" section as suggested, see P4L94-95.

3. P7L178: Should the confidence score read 9.2?

As Equation 6 indicates in P8L192, a floor function $\lfloor x \rfloor$ is applied to the product, which gives as output the greatest integer less than or equal to x .

$$\text{confidence score (CS)} = \lfloor 10 \times (p_1 - p_2) \rfloor \text{ (Equation 6)}$$

For example: $\lfloor 10 \times (0.95 - 0.03) \rfloor = \lfloor 10 \times (0.92) \rfloor = \lfloor 9.2 \rfloor = 9$

4. P8L182: Remove comma after "accuracy".

Comma removed as suggested.

5. P8L202: Here, the superimposed colours of ANN and "ANN random" are too similar. Moreover, this representation hinders the interpretation of the whiskers.

We decided to stick with the original colour palette, as the colours represent the same model (ANN or KNN) and the different shades the different training of them, i.e., trained on real or randomised data.

Instead of showing a box for each of the folds, accuracies across folds are aggregated on the same box. This allows a box for each method (100 data points each) to be shown,

resolving the previously difficult interpretation of the whiskers. This new layout also allowed for the addition of completely random prediction ($p=0.25$). See new Figure 3 on P9L201.

6. P11L272: I suggest clarifying the expression "more depleted" as its meaning is not clear here.

Text has been edited to explain the concept of "more depleted". See P12L290-292.

7. P11L280: ME  MES.

Fixed.

8. P15L337: I suggest adding a reference for the statement "buried residues tend to be evolutionarily conserved".

Two relevant references have been added to back up this statement:

63. Chothia, C. and A.M. Lesk, *The relation between the divergence of sequence and structure in proteins. EMBO J, 1986. 5(4): p. 823-6.*

64. Russell, R.B. and G.J. Barton, *Structural features can be unconserved in proteins with similar folds. An analysis of side-chain to side-chain contacts secondary structure and accessibility. J Mol Biol, 1994. 244(3): p. 332-50.*

9. P16L363: Are there not seventeen remaining sites?

Fixed.

10. P17L380: Remove first occurrence of the word "identified".

Fixed.

Reviewer #2 (Remarks to the Author):

General Assessment of work:

This is an appreciable effort to design a methodology to cluster ligands by binding sites using machine learning techniques. The authors' efforts are praiseworthy, and their study has the promise to offer a valuable addition to the field of studying drug target interactions.

Comments for revision:

1. The abstract should begin with an introductory sentence to provide context and highlight the importance of the research.

*We agree with the comment and have now edited the abstract to put our work into context and highlight the relevance of our research. See abstract in **P2L13-14**.*

Additionally, the sentence mentioning "An artificial neural network and a K-nearest..." is vague. It's worth noting that the experiment actually involved a multi-layer perceptron (MLP). Please be concise.

*Agreed, text edited in abstract **P2L23**.*

2. The problem definition lacks clarity. When reading the abstract, it initially appears to be addressing an unsupervised clustering problem. However, the abstract concludes by mentioning accuracy as an evaluation metric, which is not typically applicable in clustering scenarios.

We feel that the confusion possibly stems from the paper having three components. (1) identifying binding sites from interactions; (2) clustering these sites by unsupervised learning (K-means) into four robust classes and (3) predicting which of the four classes a new site

belongs to by a supervised method (MLP). We have revised the end of the introduction to clarify these points. (P3L51-52, P3L57-59 of revised manuscript).

Clustering quality is more accurately assessed using metrics like inertia, silhouette score, Davies-Bouldin index, Calinski-Harabasz index, Dunn index, Rand index, adjusted Rand index, normalized mutual information, and others specifically designed for this purpose. Interestingly, none of these metrics are mentioned in the results section.

We agree these are all good metrics to apply. The quality of the clustering is indeed assessed by two of the methods suggested: the elbow and silhouette method, as well as a comparison with hierarchical clustering, dimensionality reduction (MDS) and bootstrapping (P6L146-148).

We have now also calculated Calinski-Harabasz index (CHI), and the Davies-Bouldin Index (DBI) as suggested, and the results agree with our previous clustering quality assessment using the above-mentioned metrics. The four metrics: Inertia, Silhouette, CHI, and DBI, agree the optimal clustering lies in 4-6 clusters. After contextualising these results with the MDS visualisation and the Ward hierarchical clustering, the decision of $K = 4$ is further supported.

We have revised the manuscript accordingly in P6L151-153. Figure 2 added in supplementary and referenced in P6L154.

Again, when we read about the design of ANN model in the “Binding site cluster prediction” section, it seems like this is a supervised classification problem, given that the output layer consists of 4 output nodes (one for each cluster label). Things are not organized properly, please be clear.

As explained above, we have added text in the Introduction to clarify the steps in our analysis. In the final step of “Binding site cluster prediction” a multilayer perceptron (MLP) is trained on the labels obtained via the K-means clustering and labels are predicted in a supervised manner (P7L166-167).

Adjusted Mutual Information (AMI) as well as Adjusted Rand Index (ARI) have been calculated to further support the performance of the MLP. Text edited in P9L226-230.

3. Relying solely on Accuracy is an inadequate metric for assessment; it is advisable to incorporate additional metrics for a more comprehensive evaluation of your work.

We have now calculated the Adjusted Mutual Information (AMI) and Adjusted Rand Index (ARI) to further assess the performance of the MLP classifier. The text has been revised accordingly in P9L226-230.

4. In the 'Binding site definition' section, provide a comprehensive overview of drug-target interactions, considering that factors such as binding affinity and binding pose (structural characteristics) are equally crucial for comprehending the binding site. You may refer to the referenced paper for guidance: (<https://doi.org/10.1093/bib/bbab476>).

While it is true that binding affinity, as well as poses are important features to consider, our method aims to group ligands into distinct binding sites on the surface of the protein, based on their protein-ligand interactions. For this purpose, simply looking at the ligand-interacting residues is enough to be able to group the ligands together into different sites.

Indeed, for further stratification into sub-site/pockets, data such as affinity or the type of the interactions would be important. This is something we plan to explore further in our next work.

5. I am sceptical about the phrase: 'These results suggest that the KNN is better than the ANN at predicting RSA-based cluster labels' in the section 'Predicting RSA cluster membership'. It's worth noting that the 'ANN' mentioned in this context employs only a single hidden layer, making it the most basic form of a neural network. Therefore, it would not be appropriate to draw a conclusion about KNN's superiority over ANN without conducting any ablation studies or further analysis.

*These are valid points. We are not concluding KNN is superior to ANN as a method in general, but it does seem to be the case in the cross-validation (CV) of these two KNN and ANN models. It is true that we are employing almost the simplest instance of an MLP with a single hidden layer, but its CV accuracy is already $\approx 90\%$ (Figure 3, **P8L201**).*

*As mentioned in **P8L193**, this difference is likely a consequence of the different input data that goes into each model. KNN uses the U_D matrix calculated directly from the RSA profiles. Accordingly, each data point is represented by a 293-element vector containing the distance to every other binding site on the dataset. For the ANN, sites are encapsulated as 11-element vectors to make all site vectors the same length. No information relative to the other sites is present in here and probably leading to the slightly inferior accuracy of the MLP.*

*When testing both models on the blind set, there is no significant difference between them, and that is the conclusion. To clarify these points, the manuscript has been substantially revised in **P8L193-196**.*

6. Continuing the comment #5, Authors need to perform various ablation studies as mentioned below:

We agree that all the suggestions made are standard in developing a machine learning approach. Although this hyperparameter optimisation had already been explored, leading to the MLP presented in the manuscript, we had not done this as systematically as the reviewer suggested. For this reason, we thank the reviewer for their insightful suggestion, as it helped us better understand our model. The manuscript has been revised in **P8L179-181** to explain our findings, and Supplementary Figure 3 added, and referenced on **P8L181**.

The figure above represents the effect that each hyperparameter change has on the prediction accuracy relative to our current ML setup, labelled as **current**. Box and whiskers represent the distribution of validation accuracies across 100 random seeds. Dashed lines mark the separation between different hyperparameters, e.g., number of layers, neurons, activation, loss functions, etc.

To answer these questions, a series of ablation studies were carried out. Sixty-four single-hyperparameter changes were performed, one at a time. For each variation, 100 models were trained with different seeds and the average validation accuracies compared to our current MLP. Sixty-four pairwise t-tests were conducted to compare the accuracy means, and Benjamini-Hochberg correction applied. We will use FDR and $\Delta_{acc} = acc_{VARIANT} - acc_{CURRENT}$ used to describe the results, where $acc_{CURRENT}$ is the average validation accuracy of our current ML setup across the 100 seeds, and $acc_{VARIANT}$ is the average accuracy across 100 seeds of each one of the 64 variant models.

$\Delta_{acc} < 0$ will represent a decrease in performance respect our current ML architecture, whereas $\Delta_{acc} > 0$ will mean a higher accuracy.

a) Layer removal: Remove one or more layers (e.g., hidden layers) from the MLP and observe how it affects the model's performance. This helps assess the necessity of each layer in the network.

Removing the single hidden layer resulted in a significant decrease in accuracy, $\Delta_{acc} = -11\%$ ($FDR < 0.05$).

The addition of more layers did not improve accuracy: 2-layer $\Delta_{acc} = -1\%$ ($FDR < 0.05$), 10-layer $\Delta_{acc} = -8.9\%$ ($FDR < 0.05$), or was not statistically different from our current setup baseline: 5-layer $\Delta_{acc} = -0.15\%$ ($FDR = 0.42$).

b) Node Removal: Experiment by selectively removing neurons (nodes) from one or more layers to evaluate the impact on model performance.

The addition of neurons $N_{neurons} = [11, 20, 25, 50, 100]$ in the single layer did not improve the current accuracy ($FDR > 0.05$).

The removal of neurons did not have an effect of performance $N_{neurons} = [4, 5, 6, 7, 8, 9]$ ($FDR > 0.05$), or a significant negative effect for 1 neuron, $\Delta_{acc} = -15\%$ ($FDR < 0.05$), 2 neurons $\Delta_{acc} = -4\%$ ($FDR < 0.05$), and 3 neurons, $\Delta_{acc} = -1\%$ ($FDR < 0.05$).

This result suggests that 5 neurons on a single hidden layer might be enough to achieve a comparable accuracy to our current model.

c) Activation Function Ablation: Assess the effect of using different activation functions in the MLP. For example, you can compare results when using ReLU, sigmoid, or tanh activation functions in various layers.

The usage of different activation functions either negatively affected the accuracy of the MLP ($\Delta_{acc} < 0$) or had no effect ($FDR > 0.05$).

d) Weight Initialization: Investigate the influence of different weight initialization methods, such as random initialization, Xavier/Glorot initialization, or He initialization. Ablate certain weight initialization techniques to see how they affect convergence and model performance.

Most weight initialisers were tested and either negatively affected the accuracy of the MLP ($\Delta_{acc} < 0$) or had no effect ($FDR > 0.05$). However, RandomNormal, RandomUniform, and TruncatedNormal did improve the accuracy but by less than 1%, $\Delta_{acc} < +1\%$, ($FDR < 0.05$).

e) Hyperparameter Ablation: Systematically change and ablate various hyperparameters like the number of hidden units, the number of layers, or the dropout rate to understand their impact on the model's performance.

The effect of adding/removing layers has been discussed in a) and see b) for the addition/removal of units.

Regarding dropout rates, a rate = 75%, negatively affected prediction $\Delta_{acc} < -2\%$, ($FDR < 0.05$). Lower dropout rates: 0.1, 0.25, and 0.33 did improve the accuracy, but the effect size is very small, $\Delta_{acc} < +1\%$, ($FDR < 0.05$).

This result agrees with b) and shows that fewer neurons on a single hidden layer might be enough to achieve a comparable accuracy to our current model, as dropping them out has no effect.

f) Loss Function Variations: Experiment with different loss functions (e.g., mean squared error, cross-entropy) to see how they affect the training and performance of the MLP.

Different loss functions resulted in terrible loss of accuracy $\Delta_{acc} \approx -50\%$, ($FDR < 0.05$). This is expected as they are not appropriate for a multi-label classifier, unlike sparse categorical cross entropy.

Additionally, you have the option to conduct ablation experiments involving regularization techniques, learning rates, and optimization algorithms. Create a table to document all the outcomes and subsequently compare these results with those from the KNN approach.

Regarding optimisers, they either severely negatively affected accuracy $\Delta_{acc} \approx -30\%$, ($FDR < 0.05$), had no significant effect ($FDR > 0.05$), or very slightly improved accuracy, such as RMSProp $\Delta_{acc} < +1\%$, ($FDR < 0.05$).

Extreme learning rates of 0.001 (too small), and 1.0 (too big) negatively affected prediction $\Delta_{acc} < -5\%$, ($FDR < 0.05$). Intermediate rates had either no significant effect ($FDR < 0.05$) nor relevant $|\Delta_{acc}| < 1\%$.

Overall, implementing kernel, bias, or activity regularisation techniques did not improve prediction accuracy, but worsened it $\Delta_{acc} \in [-2.56, -0.46]$, ($FDR < 0.05$).

7. Ensure that the readme documentation in the GitHub repository is comprehensive and clear enough to guide anyone attempting to replicate the results. Presently, the instructions for reproducing the results are insufficient.

We appreciate the feedback on this and have made some changes that hopefully improve clarity.

Reviewer #3 (Remarks to the Author):

The authors discuss a novel method to elucidate binding site identification. There are several fragment/solvent-based MD simulations (MDMix, MixMD, SILCS) that have successfully been used for site IDing.

1. The authors should discuss the validity of these methods and preface them in the INTRO Section.

MDMix/MixMD: Molecular Dynamics simulations with MIXed solvents.

SILCS: Site Identification by Ligand Competitive Saturation. This is a computational method that uses molecular dynamics (MD) simulations of the target protein in solution with a variety of small molecules. After multiple replicates of the simulation, ligand trajectories are combined, probability maps calculated, and normalised, and protein-ligand binding energies calculated.

While all these methods are of great value, and well recognised and used in the scientific community, they tackle a slightly different problem to the one that is approached in this manuscript. These methods identify, or predict sites by using computational methods, i.e., molecular dynamics. They simulate ligand trajectories and estimate interaction energies.

In this work, we are working with hundreds of experimentally determined 3D structures of protein-ligand complexes. The structures show where the ligands interact with the protein. What we do is group the different molecules binding to the same protein with the aim of being able to discern different locations of the protein surface where different ligands bind, based on the protein residues they interact with, so these distinct sites can be characterised and analysed downstream.

*To clarify the difference between what these quality methods do and what we are doing on this work, the manuscript has been edited and appropriate references added in **P2L41-P3L45**.*

2. What makes this method unique should be clearly articulated.

*We feel that we have explained the uniqueness of our approach in paragraph 3 of the introduction. However, we have revised the text at the end of paragraph 3 further to clarify the steps in our approach (**P3L51-52, P3L57-59**).*

3. Thirdly the authors must compare their data to at least freely accessible FTMap tool and analyse how the results differ from their method versus FTMap, and if any analyses could be shown to prove their findings.

As we explain in our response to the first comment by Reviewer #3, our method is not predicting binding sites but defining them systematically from experimental data. Accordingly, a comparison with methods that predict sites is outside the scope of this paper.

*Predictions could certainly be used as input to our method in place of experimentally determined structures. This is something we are investigating as explained in the Discussion, **P23L514-519**.*

4. The functional validation of sites is an interesting concept, it would be useful if the authors could perform a quick case study on MPro (the main protease for COVID-19) which has 2 allosteric sites and then show how their method can predict the allosteric communication. Please see paper  <https://pubmed.ncbi.nlm.nih.gov/35107014/> and try to analyse if these allosteric sites are indeed predicted through this method.

We appreciate this suggestion of a system that was not used in developing our methods. For MPro we gathered 971 ligands, 465 unique, from 511 structures deposited on the PDB, most of them from fragment screening experiments. These ligands were grouped into 25 different ligand binding sites. The majority of the ligands, 891 (92%) were bound to the active site.

However, the remaining 8% bound to a diverse range of sites around the dimer surface, and pockets. Of the 25 sites, 8 were classed as C1, 12 as C2, 3 as C3 and only 2 as C4. Of the 8 C1 sites, B50 corresponds to the active site and sites 8, 11, and 19 correspond to allosteric sites 1, 2, and 3 [1] respectively. B514 is at the interface between the two copies of the enzyme and known to be a potential allosteric site. The remaining three sites present only a single ligand binding to them.

A

B

This analysis and result are now included in the Discussion section of the manuscript P23L507-513. Supplementary Figure 4 has been added and referenced in the manuscript in P22L511.

1. *DasGupta D, Chan WKB, Carlson HA. Computational Identification of Possible Allosteric Sites and Modulators of the SARS-CoV-2 Main Protease. J Chem Inf Model. 2022 Feb 14;62(3):618-626. doi: 10.1021/acs.jcim.1c01223. Epub 2022 Feb 2. PMID: 35107014; PMCID: PMC10262278.*

REVIEWERS' COMMENTS:

Reviewer #1 (Remarks to the Author):

I believe the authors fully satisfied all the suggested modifications. The final work is improved and I believe it will have a positive impact on the community.

Reviewer #2 (Remarks to the Author):

Although I am generally satisfied with the revision, I request the authors to organize the results of hyperparameter tuning in a tabular format. Additionally, please include the findings of the ablation study in the supplementary notes.

The GitHub is still very poor and complicated. First thing, there is no instruction to clone the project, download trained models, download test data.

Second, there are no instructions to setup the environment using Anaconda or Docker. I would suggest using Anaconda in your case. Please make sure that you don't just list the dependencies, rather write instructions to create a conda environment using the .yml file to install all the requirements and dependencies.

Again, there are not sufficient instructions of codes to run in order to reproduce the train and test results.

Reviewer #3 (Remarks to the Author):

Liked the edits and the revision.
Thanks for clarifying my comments.

Response to reviewers

Reviewers #1 and #3 were both happy with our revisions and had no further comments.

Reviewer #2 had some further comments: 1 to express the hyperparameter optimisation as a Table and also to make revisions to the GitHub repository. We have answered both these minor suggestions below.

Reviewer #2 (Remarks to the Author):

Although I am generally satisfied with the revision, I request the authors to organize the results of hyperparameter tuning in a tabular format. Additionally, please include the findings of the ablation study in the supplementary notes.

The GitHub is still very poor and complicated. First thing, there is no instruction to clone the project, download trained models, download test data.

Second, there are no instructions to setup the environment using Anaconda or Docker. I would suggest using Anaconda in your case. Please make sure that you don't just list the dependencies, rather write instructions to create a conda environment using the .yaml file to install all the requirements and dependencies.

Again, there are not sufficient instructions of codes to run in order to reproduce the train and test results.

We have now added Supplementary Table 1 in addition to Supplementary Figure 3 which includes the accuracy of each variant MLP model, the accuracy difference relative to our baseline ML setup, and the FDR of that difference (P6). Additionally, we have added Supplementary Note 1 to describe the MLP ablation studies.

A new **INSTALL** file has been added to the repository which includes all the necessary instructions to clone the repository and install all necessary software from .yml files.

See <https://github.com/bartongroup/FRAGSYS/blob/main/INSTALL.md> for reference. This is linked to from the README <https://github.com/bartongroup/FRAGSYS/blob/main/README.md>

An extra notebook has been added to illustrate how one can use the MLP to predict RSA Cluster labels from binding site RSA profiles.

See https://github.com/bartongroup/FRAGSYS/blob/main/analysis/15_ML_predicting_rsa_labels.ipynb for reference.